# Evoked temporal summation in dogs to assess pain central sensitization and modulation – A feasibility study

Aliénor Delsart[1], Maude Barbeau-Grégoire[1], Maxim Moreau[1,2], Colombe Otis[1], Aude Castel[1,3], Bertrand Lussier[1,2,3], Johanne Martel-Pelletier[1,2], Jean-Pierre Pelletier[1,2], Eric Troncy[1,2]*

1 Groupe de Recherche en Pharmacologie Animale du Québec (GREPAQ), Université de Montréal, Quebec, Canada, 2 Osteoarthritis Research Unit, University of Montreal Hospital Research Center (CRCHUM), Quebec, Canada, 3 Department of Clinical Sciences, Faculté de Médecine Vétérinaire, Université de Montréal, Quebec, Canada

* eric.troncy@umontreal.ca

## Abstract

Central sensitization and pain endogenous controls imbalance were reported in humans affected by chronic pain. In cats, nociplastic changes, such as the wind-up phenomenon, (*i.e.*, decreased tolerance to repeated stimuli) were described using a validated mechanical device inducing temporal summation of pain (TSP). The study aim was to validate this method in dogs and to describe pro- or anti-nociceptive profiles occurring with canine osteoarthritis (OA). Healthy ($N=4$) and OA ($N=31$) dogs were assessed. Six TSP stimulation protocols were tested for their reliability and specificity, at mid-antebrachium (0.38 Hz at 2–4Newtons [N], 0.25 Hz or 0.50 Hz at 4N) or tail-base (0.25 Hz or 0.50 Hz at 2N). Endogenous facilitatory, or inhibitory, controls were assessed using a pressure pain threshold (PPT) before and after TSP, or conditioning pain modulation (CPM) *via* an ischemic model, respectively. The PPT pre/post stimuli were used to calculate ratio of facilitation (< 0%) or functional inhibition (> 7%). Statistical analyses included intraclass coefficient of correlation, Spearman's correlations, Fisher and Mann $U$ tests, with $\alpha=0.05$. The response to mechanical TSP scores tended to be moderately reliable, while tail-base stimulation decreased inter-trials variability compared to mid-antebrachium for healthy dogs ($P=0.029$). The response was specific, OA dogs tolerated 50% less stimulations than healthy dogs at 0.5 Hz frequency and 2N ($P=0.008$). This protocol led to spinal hyperexcitability; among facilitated OA dogs, 45% presented functional inhibition *versus* 73% for the non-facilitated dogs. No correlations were found between radiographic score, orthopedic score, response to mechanical TSP or age with the facilitation or inhibition ratios ($P>0.173$). OA dogs developed central sensitization reflected by the wind-up phenomenon. An imbalance in favor of endogenous facilitatory controls was observed suggesting inhibitory control fatigue and neuroplasticity.

**Data availability statement:** The datasets presented in this study can be found in an online repository: https://data.mendeley.com/datasets/np4x38hkgj/1.

**Funding:** The authors declare financial support was received for the research, authorship, and/or publication of this article. The GREPAQ® was supported by Discovery grants (RGPIN #441651-2013 and #05512-2020; E.TR.) for salaries, and Collaborative Research and Development grants (RDCPJ #418399-2011 and #491953-2016; E.TR. in partnership with ArthroLab, Inc.) for operations and salaries, both from the Natural Sciences and Engineering Research Council of Canada (NSERC). Additional support was provided through a New Opportunities Fund grant (#9483; E.TR.) and a Leader Opportunity Fund grant (#24601; E.TR.) from the Canada Foundation for Innovation (CFI), which funded pain/function equipment. C.OT. was the recipient of a MITACS Canada Postdoctoral Fellowship Elevation (IT #11643) and A.DE. the recipient of a Fonds de recherche Québec – Nature et Technologies (FRQNT) doctoral grant (#350507).

**Competing interests:** The authors have declared that no competing interests exist.

Characterizing endogenous pain modulation will enable better management of OA pain by preserving inhibitory control and preventing its fatigue.

## Introduction

Quantitative sensory testing (QST) allows the characterization of somatosensorial changes (*i.e.,* gain or loss of function) occurring during chronic pain and notably osteoarthritis (OA) [1,2]. They were developed in humans and their translation to rat model of induced OA pain and naturally occurring feline and canine OA was recently summarized [3]. Individuals affected by OA can develop peripheral (*i.e.*, decrease in excitatory threshold and amplification in the responsiveness of nociceptors) and central (*i.e.,* spontaneous activity, decrease in activation threshold by peripheral stimuli and enlarge receptive fields of dorsal horn neurons) sensitization [4–7]. An early nociplastic change characterizing central sensitization is the wind-up phenomenon. It corresponds to a decrease in tolerance to a noxious or non-noxious repeated stimulus at a specific frequency by recruiting C-fibers and Aβ-fibers during maladaptive pain [8]. It led to an increase in pain perception despite no change in stimulus magnitude, reflecting temporal summation. The wind-up remains reversible contrary to the long-term potentiation and allows the discrimination between healthy and neuro-sensitized human patients [4]. In cats, this phenomenon was characterized using the response to mechanical temporal summation of pain (TSP) with a device that delivers mechanical stimulations under the nociceptive threshold [9,10]. It was validated for its reliability, specificity (*i.e.*, able to distinguish between healthy and OA cats), and sensitivity (*i.e.*, responsive to tramadol but not meloxicam) [6,10–12]. The only studies assessing TSP in dogs used an electrodiagnostic test, mostly performed under anesthesia, to elicit a nociceptive withdrawal reflex [13–15]. The response was recorded by an electromyogram, and the limb flexion was scored. However, this is not a QST and did not reflect the conscious perception of pain including the motive-affective component, reflected by aversive behavioral responses. To date, no studies have been performed with mechanical stimuli in conscious dogs to assess spinal hyperexcitability. The validation of a method for assessing spinal central sensitization in conscious OA dogs is necessary to better characterize this degenerative disease.

The perception of OA pain results from endogenous modulation that can be illustrated by the facilitation and inhibition (F/I) balance, *via* descending or local pathways [16,17]. Facilitation was described using TSP studies in humans, with a higher pain ratio at the end of the series of stimulations compared to baseline [18]. On the other hand, the diffuse noxious inhibitory control (DNIC) first described in rats, refers to the "pain inhibits pain" phenomenon [19]. It was later replaced by the conditioned pain modulation (CPM) appellation to integrate behavioral correlate presents in humans' studies and proposed similarly to companion animals [20–22]. The use of mechanical or ischemic pain as conditioning stimulus distant to the primary test stimulus was validated for assessing CPM in dogs [20,21]. Several phenotypes were determined in human patients affected by chronic pain, such as enhanced TSP and/or dysfunctional CPM (*i.e.*,

pro-nociceptive) or not (*i.e.*, anti-nociceptive) [17,23]. The translation of these pain endogenous control profiles described in humans to animals was discussed in a recent review, and some hypotheses were emitted for feline and canine OA, but remained to be confirmed [3]. In 2020 a metanalysis concluded that OA cats were peripherally and centrally sensitized compared to healthy, suggesting a pro-nociceptive profile, even if F/I balance remains to be characterized [6]. In OA dogs, the lack of homogeneity in data acquisition across studies (difference in anatomical locations, experimental design and few studies with both OA and healthy dogs) prevented metanalysis [7]. Standardized and validated QST protocols are needed; only one guidelines' article has been published but this does not include the neuromodulation assessment [24]. However, in distinct studies including different cohorts of dogs, sensitization was described at the local affected joint and at remote sites, corroborating a generalized state of neuro-sensitization, as well as a dysfunctional CPM for OA dogs [21,25–27]. The suggested central sensitization was supported by changes in spinal cord biomarkers (substance P and transthyretin) in an experimental model of canine OA, as it was clearly validated in an experimental OA model in rats [28,29]. There is a need to validate tools for assessing pain endogenous neuromodulation in dogs and to describe the F/I balance during OA.

The aims of this study were to validate the response to mechanical TSP for its specificity to distinguish healthy dogs from dogs with OA, for its sensitivity regarding the level of impairment and facilitation, and to determine both pro- and anti-nociceptive endogenous neuromodulation profiles. The hypotheses were: (1) the response to mechanical TSP will be reliable in-between trials, and OA dogs will present central sensitization reflected by a reduced number of accepted stimuli compared to healthy dogs (specificity), (2) an F/I imbalance will be reported in OA dogs and (3) neuro-sensitization profiles will correlate with OA functional alterations.

## Materials and methods

### Ethics statement

The study was approved by the Institutional Animal Care and Use Committee of ArthroLab Inc. (A197-ART22D). Care of animals adhered to the regulations outlined in the Canadian Council on Animal Care Guide to the Care and Use of Experimental Animals, Vol. 1, 2nd Edition (1993, revised in Feb. 2017), and complied with the US Dept. of Agriculture Animal Welfare Act (9 CFR Parts 1–3). All procedures also adhered to the ARRIVE guidelines for reporting animal research and the Committee for Research Ethical Issues of the International Association for the Study of Pain guidelines.

### Animals

Adult neutered dogs, healthy ($N = 4$) or with naturally occurring OA ($N = 31$) were included in this study. The selection of OA dogs was based on radiographic evidence of OA in at least one joint, confirmed by an orthopedic exam and a deficit in one pelvic limb on the static weight bearing analysis. Four radiographic images were collected (ventro-dorsal view of the pelvis, lateral views of each stifle and lateral view of both tarsus) and scored (0–3) for the following criteria, presence of osteophytes, subchondral sclerosis and joint effusion at each joint. All evaluations were performed by a board-certified veterinary surgeon (B.LU.). Apart from their OA status, dogs were healthy according to physical examination (weight and body condition) and absence of clinical pathological findings based on blood sample analysis (complete blood counts and serum chemistry) and urinalyses (when required by the veterinarian). All dogs were selected based on their compliant behavior and their ability to respond to commands (to sit and lay down).

All animals were housed in pairs in a kennel facility with wire enclosures. Their health status was monitored daily by the local medical team. The dogs were fed with standard certified commercial diet (Pronature® Original Adult® All Breeds, Pronature, Inc., QC, Canada; Purina® OM® Overweight Management Canine Formulas, Purina, Inc., MO, United States; Hill's® Prescription Diet i/d®, Hill's Pet Nutrition, Inc., ON, Canada) according to the manufacturers or the veterinarian's recommendations. Fresh water was provided *ad libitum*.

## Experimental design

Prior to the beginning of the study, all animals were submitted to an acclimation phase to avoid stress-induced analgesia, for six acclimation sessions of 10–15 min over three weeks [30]. Dogs were encouraged to sit or lay-down in a foam bed and gradually acclimated to the female evaluators, the assessment room and habituated to the device (including the armband, noise of the stimulating system and the stimulation at a halved intensity). All evaluators were blinded to the dog's health status.

To determine which TSP protocol would be the most adapted, the reliability of the different protocols was first tested on healthy dogs, varying the intensity, the frequency and the location of the stimulation (mid-antebrachium *vs*. tail-base); second the specificity was tested for most reliable protocols in both locations, by comparing the results obtained for mechanical TSP measurement between healthy and OA dogs (Table 1). For the F/I neuromodulation assessment, the effects of TSP (selected from the first part of the study) and CPM were quantified using a Pre/Post measure of pressure pain threshold (PPT). Assessments were performed on separate weeks to test for the repeatability of measurements within time. Details of the procedures and experimental design are presented below. The same healthy dogs were included in all stimulation protocols, except for one dog during the tail-base stimulations (Protocols E and F).

## Response to mechanical temporal summation of pain

Dogs were encouraged to lay down or sit in a foam bed (Fig 1) with a minimal restraint (only if necessary). Once calm, the repeated mechanical stimuli were applied using a specific device (Top Cat®, Bespoke Measurement Systems, Cambs, UK). Criteria of mechanical TSP induction by the device include the intensity (in Newtons, N), the frequency (in Hertz, Hz) and the stimulation duration (sec) of the repeated stimuli (Table 1).

The non-noxious mechanical stimulus was delivered using a 10 mm long metal rod with a 2.5 mm diameter rounded tip mounted on a rolling diaphragm actuator and held using an armband. For the mid-antebrachium stimulation assay, a dummy device (an actuator without the metal pin) was placed on the contralateral limb (left mid-antebrachium) as a negative control. The stimulation consisted of the pin moving back and forth perpendicularly to the skin, and it was stopped when two consistent aversive reactions were observed (dogs vocalizes, licks the cuff, withdraws the paw, stands up, *etc*.) or the maximum of 30 stimulations was reached. A 4-minute break was then completed before the second stimulation session. If any aversive reaction was linked to an environmental stimulus (light, sound, *etc*.), the procedure was stopped and resumed after a 4-minute break to get a reliable reaction. A reliability score, based on a Likert scale, was associated with the number of stimulations tolerated (2: very reliable, 1: moderately reliable, 0: not reliable). Decreases in the number of tolerated stimulations were considered to reflect increase in spinal sensitization. In the absence of any "duplicate effect", the mechanical TSP averaged values were weighted with the reliability numbers at each trial (*i.e*., with duplicate values)

**Table 1. Temporal summation stimulation protocols description.**

|  | Protocols | Number of dogs | Intensity (Newtons, N) | Frequency (Hz) | Stimulation duration (sec) | Inter-stimulation pause (s) |
|---|---|---|---|---|---|---|
| Mid-antebrachial stimulations | A | OA: 28; Healthy: 4 | 2 | 0.38 | 1.3 | 1.3 |
|  | B | OA: 28; Healthy: 4 | 4 | 0.38 | 1.3 | 1.3 |
|  | C | Healthy: 4 | 4 | 0.25 | 1.3 | 2.7 |
|  | D | Healthy: 4 | 4 | 0.50 | 1.0 | 1.0 |
| Base of the tail stimulations | E | Healthy: 4 | 2 | 0.25 | 1.3 | 2.7 |
|  | F | OA: 31; Healthy: 4 | 2 | 0.50 | 1.0 | 1.0 |

OA: Osteoarthritis.

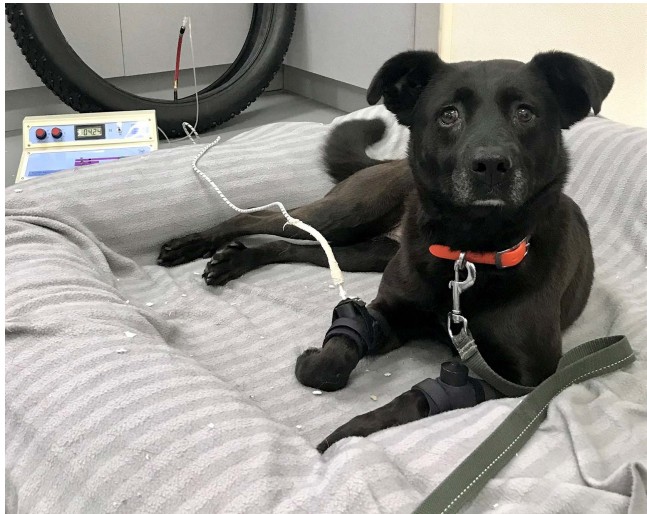

**Fig 1. Photograph of the mechanical temporal summation set-up when performed at the mid-antebrachium.** Dogs were laying down on a comfortable cushion. The mechanical stimulator was placed at the mid-antebrachium (right) and connected to the stimulator device. A dummy cuff was placed at the contralateral forearm (left). The photo was gratiously provided by the GREPAQ® with the permission to the open-access journal PLoS-ONE to publish it under a CC BY license.

and used for analysis. Each protocols A to F were tested within three trials (except for protocol F in OA dogs, which was tested with two trials), with a one-week delay between each trial (Fig 2).

## Pain endogenous neuromodulation

The pain endogenous neuromodulation was assessed as previously described [3], using either the selected mechanical TSP protocol for best reliability/specificity (*i.e.*, facilitation neuromodulation) or CPM assessment (*i.e.*, inhibition neuromodulation). Briefly, a duplicate measure of PPT was determined in N using an algometer (Wagner Force Gauge M3-2, Mark-10 Corporation, Copiague, NY, USA). The PPTs were taken before and after either the mechanical TSP duplicate sessions or the single CPM assessment using an ischemic pain model (with a manometric cuff inducing vascular blockade) [20] in the following minute. To record PPT, the pressure was gradually increased (2.5N/s) at the metatarsus (perpendicular orientation) of the most affected limb until an aversive response was observed (paw withdrawal, agitation, weight shift, vocalization, licking, *etc*.) or the maximum of 10N for PPT was reached.

Facilitation and inhibition trials were recorded twice, both on separate days, one week apart. The facilitation or inhibition ratio was calculated following Formula (1). A negative value with TSP meant facilitation, whereas a functional inhibition with CPM was determined when the percentage of change was positive and greater than 7% (internal data 2021, healthy dogs median ratio of 7%) or a maximum threshold (10N) was reached (both pre- and post-stimulus, or either pre- or post-stimulus with the second threshold less than 7% close).

$$Facilitation\ or\ inhibition\ ratio = \frac{(Post\ Stimulus - Pre\ Stimulus)}{Pre\ Stimulus} \times 100 \tag{1}$$

## Statistical analyses

Descriptive analyses were performed with the median (min-max) and comparisons between protocols and groups were determined using non-parametric Mann-Whitney $U$ test, Fisher exact or chi-square ($\chi^2$) test. The coefficient of

**Fig 2. Timeline of the experimental design.** See Table 1 for each protocol description.

dispersion (COD) was used to describe the dispersion of the data, expressed in percentage (%) and calculated using the formula (2):

$$COD = \frac{\frac{\sum |xi-Median|}{Sample\ size}}{Median} \qquad (2)$$

For the inter-trials COD, $xi$ was the value for each trial and for the inter-individuals COD, $xi$ was the mean of the trials. The repeatability between trials (or duplicate) was assessed using the intraclass coefficient of correlation (ICC) and their 95% confidence intervals, using single or mean-rating, absolute agreement and 2-way mixed effects. The ICC strength was as follows: less than 0.50 corresponded to poor, between 0.50 and 0.75 to moderate, between 0.75 and 0.90 to good, and greater than 0.90 indicated excellent reliability [31]. Spearman rank test correlations (coefficient of correlation: Rho$_S$, $\rho_S$) were calculated between the intensity of facilitation or inhibition *vs.* other scores (radiographic, orthopedic or response to mechanical TSP) or age. The α threshold was set at 0.05 to be considered significant. Data was analyzed using IBM® SPSS® Statistics software (IBM Corp. Released 2023. IBM SPSS Statistics for Windows, Version 29.0.2.0 Armonk, NY: IBM Corp).

## Results

### Characteristics of the dog population

Healthy dogs presented no radiographic alteration, were younger than OA dogs ($P<0.001$) and weighed the same ($P>0.861$) (see Table 2). The most affected joint of OA dogs was the hip (75–87%). The number of females was similarly distributed between healthy and OA dogs for mid-antebrachium stimulation (Table 2), but OA dogs included more females for tail-base stimulation (probably due to the difference in sample size).

### Reliability of the response to mechanical temporal summation of pain – Healthy dogs

**Mid-antebrachium stimulations.** Intensity effect. Healthy dogs ($N=4$) were assessed with protocols A to D at mid-antebrachium (Table 1 and Table 3). In protocols A and B, stimulations applied on mid-antebrachium differed only in intensity (in N). The protocol A tended to be moderately repeatable (ICC: 0.678 [–0.365–0.977]; $P=0.084$) and the repeatability decreased for protocol B (higher intensity; ICC: 0.245 [–1.164–0.936]; $P=0.320$). However, the reliability of protocol B raised when excluding the second trial (ICC: 0.897 [–1.152–0.993]; $P=0.062$). By increasing the intensity of stimulations, the inter-trials COD decreased by 43%, and the inter-individuals COD dropped (Table 3).

Frequency effect. The higher intensity (4N) of protocol B was selected for frequency characterization. The protocol C (lower frequency) was highly reliable (ICC: 0.852 [0.277–0.989]; $P=0.011$) while the protocol D (higher frequency) tended to be repeatable (ICC: 0.694 [–0.694 [–0.682–0.979]; $P=0.101$). The number of stimulations accepted in protocols B, C and D were similar ($P>0.773$). Compared to protocol B, the inter-trials COD increased by 45% for protocol C and by 66% for protocol D. The inter-individuals COD was more than doubled for both protocols compared to protocol B (Table 3).

**Table 2. Characteristics of the dogs recruited for the mid-antebrachium and tail-base stimulations.**

|  | Mid-antebrachium stimulation | | | Tail-base stimulation | | |
|---|---|---|---|---|---|---|
|  | OA | Healthy | *P*-value | OA | Healthy | *P*-value |
| Number of dogs | 28 | 4 | – | 31 | 4 | – |
| Age (year) | 11 [3 –14] | 4 [3 –5] | < 0.001 | 10 [5 –15] | 4 [4 –5] | < 0.001 |
| Body weight (kg) | 26.0 [12.2–41.9] | 25.4 [22.7–28.3] | 0.934 | 27.0 [14.1–39.6] | 28.5 [19.4–33.4] | 0.861 |
| Sex (male/female) | 9/19 | 3/1 | 0.136 | 6/25 | 3/1 | 0.044 |
| Radiographic score | 8.5 [2.0–59.0] | 0.0 [0.0–0.0] | < 0.001 | 8.0 [1.0–63.0] | 0.0 [0.0–0.0] | < 0.001 |
| Most affected joint |  |  |  |  |  |  |
| Hip | 21 | – | – | 27 | – | – |
| Stifle | 4 | – | – | 3 | – | – |
| Tarsus | 2 | – | – | 1 | – | – |

Age, body weight and radiographic score are expressed as median [min-max]. Sex and the osteoarthritic most affected joint are reported in number of individuals. OA: Osteoarthritis. P-values were calculated with the Mann-Whitney U test (age, body weight and radiographic score) or the Fisher exact test (sex).

**Table 3. Raw data descriptive analysis of the different stimulation protocols at the mid-antebrachium or tail-base for healthy and osteoarthritic dogs.**

| Protocol | Status | Number of counts (median [min-max]) | Inter-trials COD (%) | Inter-individuals COD (%) | *P*-value (OA *vs.* Healthy) |
|---|---|---|---|---|---|
| Mid-antebrachium |  |  |  |  |  |
| A | Healthy | 16.0 [11.9–25.3] | 23.2 | 6.5 |  |
|  | OA | 9.4 [5.0–18.4] | 27.7 | 1.1 | 0.005 |
| B | Healthy | 13.2 [12.7–17.5] | 13.2 | 3.0 |  |
|  | OA | 8.8 [4.0–19.5] | 21.0 | 1.2 | 0.002 |
| C | Healthy | 13.9 [9.3–25.2] | 19.1 | 10.5 | N/A |
| D | Healthy | 14.4 [11.1–22.5] | 21.9 | 6.5 | N/A |
| Tail-base |  |  |  |  |  |
| E | Healthy | 23.3 [20.3–29.3] | 9.0 | 2.6 | N/A |
| F | Healthy | 27.3 [25.7–27.8] | 9.6 | 1.3 |  |
|  | OA | 14.0 [5.5–30.0] | 20.0 | 1.3 | 0.008 |

COD: Coefficient of dispersion; OA: Osteoarthritis.

**Tail-base stimulations.** Considering that none of the protocols tested on the mid-antebrachium was convincing, another location (the base of the tail) was evaluated (protocols E and F in Table 1). The ICCs were not significant ($P > 0.327$; probably due to a type II error). The inter-trials COD decreased by 53% for protocol E compared to protocol C (same frequency), and by 56% for protocol F compared to protocol D (same frequency). The inter-individuals COD decreased by 75% for protocol E compared to C and by 80% for protocol F compared to D. Comparing both protocols E and F, healthy dogs halved their inter-individuals COD in protocol F. The number of stimulations tolerated by healthy dogs for the same frequency but at the mid-antebrachium (protocols C and D) or tail-base (protocols E and F) was increased by 67% for the 0.25 Hz ($P = 0.149$) and by 89% for the 0.50 Hz ($P = 0.029$) at the tail-base location.

**Specificity of the response to mechanical temporal summation of pain – Osteoarthritis *versus* healthy**

**Mid-antebrachium stimulations.  Reliability.** Osteoarthritic dogs ($N=28$) were evaluated using protocols A and B, selected for their reliability and low COD when assessing healthy dogs (Tables 1 and 3). The protocol A was not repeatable (ICC: 0.249 [–0.496–0.594]; $P=0.249$) while the protocol B was moderately repeatable (ICC: 0.705 [0.444–0.856]; $P<0.001$) for OA dogs. By increasing the intensity of stimulation (Table 1; *i.e.,* protocol B compared to A), the inter-trials COD decreased by 24%, whereas the inter-individuals COD remained stable (Table 3).

**Specificity.** The OA dogs accepted a median of 41% less stimulations than healthy dogs with protocol A ($P=0.005$) and 33% less with protocol B ($P=0.002$) (Table 3).

**Tail-base stimulations.  Reliability.** Osteoarthritic dogs ($N=31$) were evaluated using protocol F, as it was deemed the most reliable for healthy dogs (Tables 1 and 3). This protocol was moderately repeatable (ICC=0.710 [0.405–0.860]; $P<0.001$) for OA dogs.

**Specificity.** Considering the protocol F, OA dogs tolerated half the number of mechanical stimulations compared to healthy ($P=0.008$) (Table 3).

**Neuromodulation – Facilitation and inhibition balance.**  Healthy ($N=4$) and OA dogs ($N=31$) were assessed for their pain endogenous F/I neuromodulation with the stimulation protocol F based on the results of previous sections (*i.e.*, reliability and specificity). The duplicate measures of PPT pre-TSP or pre-CPM were highly reliable for both trials (ICC>0.816 [0.667–0.902]; $P<0.001$). The PPT post-TSP was obtained within a median of 25.00 (17.50–53.00) seconds after the last mechanical TSP stimulation; and duplicate values were highly reliable (ICC>0.799 [0.637–0.893]; $P<0.001$). Similarly, the duplicate measures of PPT post-CPM were moderately to highly reliable (ICC>0.710 [0.485–0.844]; $P<0.001$).

Using the formula (1) to compare PPT pre- and post-stimulus, 75% of healthy dogs (median ratio of –11%) *vs.* 65% of OA dogs (median ratio of –15%) had results consistent with facilitation following TSP. Functional inhibition, using PPT post-CPM, was found in 100% of healthy dogs (median ratio of +7%), whereas it was present in only 55% of OA dogs (median ratio of +16%) (Fig 3). The number of dogs presenting either functional facilitation or inhibition remained similar between both sessions assessment ($\chi^2$ testing; $P>0.127$), therefore the mean value of these sessions was used for subsequent analysis.

The majority of healthy dogs presented facilitation ($N=3/4$) with a functional inhibition ($N=3/3$). Osteoarthritic dogs were either facilitated ($N=20/31$; 65%) with only 45% of them ($N=9/20$) having a functional inhibition, or non-facilitated

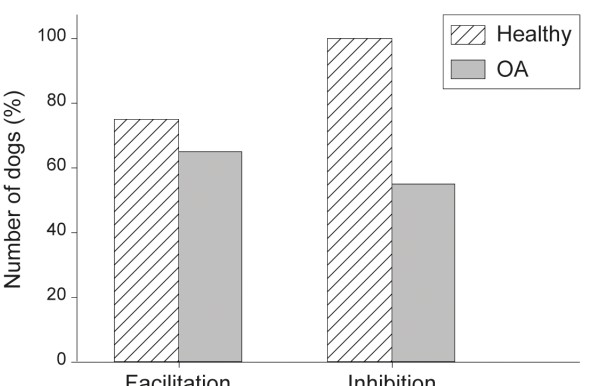

**Fig. 3.  Number of healthy and osteoarthritic dogs presenting facilitation or inhibition.** The ratio of facilitation or inhibition was calculated using Formula (1) with PPT pre- and post- TSP (facilitation) or CPM (inhibition) on both sessions of testing. For TSP, a negative value indicates that the dog presented facilitation, whereas for CPM a ratio above 7% meant functional inhibition.

(35%) with 73% (8/11) of them having a functional inhibition (ratio > +7%) (Table 4). The median intensity for facilitation of healthy dogs (–11%) was determined as the threshold to be over facilitated in OA dogs. Fourteen dogs (on 20) were highly facilitated with a median ratio of –19% compared to 12% ratio for the 11 non-facilitated ($P < 0.001$).

No significant correlations were found between either the ratio of facilitation and inhibition compared to radiographic score, orthopedic score, response to mechanical TSP score or age ($P > 0.128$). The OA dogs were classified into two groups according to their radiographic score, severely (score > 8) or mildly affected. Both groups almost halved their response to mechanical TSP score compared to healthy dogs ($P < 0.016$), suggesting an absence of relationship between the degree of structural alterations and spinal sensitization.

## Discussion

Despite 10–20% of the canine population being affected by OA, there is still a lack of assessment standardization to appropriately characterize the neuro-sensitization occurring with this degenerative disease [7,32]. Profiles according to endogenous pain modulation (i.e., F/I balance) were described in human patients and allowed to predict treatment response [18,33]. There is a need to determine these phenotypes in canine OA to improve their pain management [3].

The first step of this feasibility study was to determine the reliability according to stimulation placement (mid-antebrachium or tail-base) for healthy dogs, followed by the specificity of the response to mechanical TSP for OA dogs. The placement of the stimulus cuff was crucial. In cats it was placed and validated at mid-antebrachium with the median nerve mostly stimulated [10]. In dogs, with electrical stimulations eliciting nociceptive withdrawal reflex, the plantar digital nerve was stimulated [13,14]. As there was no mechanical stimulus reference in the literature to induce TSP in dogs, protocols tested and stimulus placement in this study were based on thus previously tested for cats with a frequency of 0.4 Hz [10]. Mid-antebrachium stimulations tended to be repeatable for healthy dogs (type II statistical error for Protocol A) and were moderately repeatable for OA dogs at higher intensity (Protocol B). For both protocols at 0.38 Hz, OA dogs tolerated significantly less stimulation than healthy dogs, whatever the intensity, supporting the specificity of the response. This was in line with physiologically findings, C-fibers spatial and temporal summation occurs with a stimulation frequency between 0.3 and 0.5 Hz [34]. This leads to magnesium ion removal of the post-synaptic NMDA-sensitive glutamate receptor, allowing glutamate mediated depolarization of the membrane and increasing the generation of action potentials in the second-order neuron [35]. Unlike cats for which either 4N or 6N was necessary to obtain specificity and discriminate between healthy and OA cats [10], an intensity of 2N or 4N was sufficient to discriminate between healthy and OA dogs. However, an intensity of 4N leads to less inter-trials variability both for healthy and OA dogs, and to less inter-individuals' variability for healthy dogs compared to a lower intensity. This was supported with higher intensities leading to less variability in cats too [10]. By comparing the number of tolerated stimulations by healthy cats and dogs, dogs

**Table 4.  Proportion of the facilitation and inhibition profiles.**

|  | Functional inhibition | Non-functional inhibition |
|---|---|---|
| Healthy dogs |  |  |
| Facilitated (< 0%) | 3 (100%) | 0 (0%) |
| Non-facilitated (≥ 0%) | 1 (100%) | 0 (0%) |
| OA dogs |  |  |
| Facilitated (< 0%) | 9 (45%) | 11 (55%) |
| Non-facilitated (≥ 0%) | 8 (73%) | 3 (27%) |

Data is expressed in the number of dogs and its associated percentage per facilitation cluster (N = 4 healthy and N = 31 OA). The ratio of facilitation or inhibition was calculated using Formula (1).

accepted 34% less stimulation. One hypothesis for the lower number of stimulations accepted by dogs was that the frequency was not optimal. Lower (0.25 Hz, Protocol C) and higher (0.50 Hz, Protocol D) frequencies were tested but resulted in more inter-trials and inter-individuals' variability for healthy dogs, these protocols corresponding to the inferior and superior limits of TSP leading to wind-up [34]. A second hypothesis to explain lower response to mechanical TSP score in dogs compared to cats, was that the mechanical stimulus was too close to the radius and induced more nociceptive stimuli. Mechanical stimuli were induced at the tail-base innervated by the caudal nerve. The protocols tested were the same as at mid-antebrachium but with a 2N intensity (a pilot study at 4N revealed intense aversive response potentially explained by nerve's ramification close to tail base). Protocols E and F were not repeatable for healthy dogs due to a statistical type II error; however, the number of stimulations tolerated by healthy dogs increased by 89% when using a frequency of 0.50 Hz and an intensity of 2N at the tail-base (Protocol F) compared to mid-antebrachium placement (Protocol D). When considering OA dogs, the reliability was moderate (close to good) and the number of tolerated stimulations halved compared to healthy individuals (see Table 3). A good to excellent reliability was described for mechanical TSP in healthy humans but was not reported for OA patients, supporting our results [36,37]. The tail-base protocol was repeatable and specific to distinguish between healthy and OA dogs validating the first hypothesis of research.

During chronic pain the concept of F/I balance was proposed in human studies [17]. Four profiles of phenotype were reported in OA human patients, either dysfunctional CPM and enhanced TSP (for 41% patients, pronociceptive phenotype), functional CPM and decreased TSP (9.4%, antinociceptive phenotype) or intermediate profiles with: Dysfunctional CPM and decreased TSP (34%), and functional CPM and enhanced TSP (16%) [23]. Although some hypotheses were emitted, no studies have already characterized profiles of pain F/I neuromodulation in OA dogs [3]. The aim of the present manuscript was to characterize this endogenous balance using mechanical TSP as a trigger to elicit facilitation and CPM using an ischemic pain model, to assess inhibition [20,21]. The response to mechanical TSP protocol F, validated at the tail-base for 0.50 Hz and 2N, induced facilitation both for healthy and OA dogs reflected by a decrease in PPT post-TSP, and with higher ratios for OA dogs. Also, the lack of difference between the duplicate post-PPT obtained within a median of 25 seconds after the end of the last TSP stimulation supports the persistence of a facilitatory process. These results confirm the inter-species nature of pain neuromodulation; indeed, in humans, cats and rats the wind-up phenomenon can last until 30–120 seconds [10,38,39]. The OA dogs cluster discrimination was similar to those of humans with two main clusters corresponding to: A pronociceptive phenotype with enhanced TSP (*i.e.*, facilitation) reported for 65% of OA dogs with mostly (55%) dysfunctional CPM, suggesting a fatigue of the pain endogenous inhibitory controls; and a decreased TSP (*i.e.*, non-facilitated, 35% of OA dogs) with functional CPM (73%), suggesting an antinociceptive phenotype. Furthermore, OA dogs with functional CPM doubled their intensity ratio compared to healthy dogs, suggesting a process to counteract facilitation. This reflects the neuroplastic changes occurring during the development of chronic pain and neuro-sensitization. In humans, different degrees of CPM impairment have been reported in OA patients, with less functional CPM associated with higher pain scores [40,41]. As in dogs, human patients with dysfunctional CPM could have increased pain sensitivity contributing to a persistent pain state reflecting a fatigue of the system [42,43]. Therefore, the second hypothesis of research was validated with changes in the F/I balance pain neuromodulation, characterized by different phenotypes in canine OA.

Given that healthy dogs were younger and that OA could develop at any age, this is an important factor to consider. Either facilitation or inhibition ratios were not correlated to age in the present manuscript. The CPM was reported to decline with age in human and rodent models, but this has not been reported in dogs [21,44,45].

No correlations were statistically significant between facilitation ratio and structural, or functional, alterations assessed by radiographic, or orthopedic scores, respectively. This is not surprising as either in dogs or in humans, no linear correlations were reported with OA functional pain and neuro-sensitization or radiographic alterations [46–48]. In cats, the response to mechanical TSP score was correlated with a static QST, the paw withdrawal threshold, but not with assessments reflecting biomechanical alterations such as peak vertical force or motor activity [10]. Moreover, a recent analysis

presented discriminatory abilities for the peripheral sensitization, such as measured with the paw withdrawal threshold, but not for the spinal sensitization (response to mechanical TSP) to the validated different degrees of OA severity [49]. These results invalid (partially) the third hypothesis of this study. The response to mechanical TSP was a marker of spinal sensitization but was not sensitive to structural alterations. However, previous studies reported the neuro-sensitization in feline OA to be reversed by tramadol (for spinal sensitization) [11,12], or by gabapentin (for peripheral sensitization) [50]. This means that when individuals develop OA, they can be centrally sensitized at any stage, not just the most advanced, as also observed in feline OA [49]. Overall, repeated mechanical stimuli under nociceptive threshold at tail-base for 0.50 Hz and 2N is recommended for measuring TSP and induced facilitation in dogs. However, the response to mechanical TSP looks like a complementary assessment method, providing information on the presence of spinal sensitization, but does not correlate linearly with functional impairment. Other QST, such as paw withdrawal threshold or the F/I neuromodulation looks more susceptible to distinguish different profiles of pain phenotype in pet animals.

Pain is a subjective experience modulated by many factors, representing limitations in the study. Indeed, aging and being a female have been reported to enhance TSP [44,51]. Dogs included in this study were all adults and the majority were senior, but aging was not correlated with decrease or enhancement of F/I neuromodulation. All individuals have been neutered, but not at the same age, which could give rise to acquired behaviors. Indeed, sexual hormone such as estrogen has demonstrated a protective role against the development of central sensitization [52], whereas early neutering (< 6 months) can influence the development of OA in dogs [53] as well as in humans [54]. Even though this is a potential bias, this reflects the real challenge in clinics. It was recently reported that pain tolerance could have varied according to dog breed, but only healthy individuals were tested, none were painful or neuro-sensitized [55]. Evaluators were aware of this phenomenon and evaluated all dogs, regardless of breed, with equal attention. To limit the impact of these age and sex factors, groups tended to be similar for stimulation of the mid-antebrachium and the tail-base. Another limit could be behavioral laterality, as dogs were assessed for TSP always on the right mid-antebrachium. Comparing TSP response from both the right and left mid-antebrachium, could help determine whether laterality influences pain perception and *vice-versa*, as is it the case with stress [56]. It must be noted that the laterality of TSP induction was tested in cats, and it was not present for right and left mid-antebrachium [10]. Chronic OA pain was previously associated with a decreased quality of life [57], it could be of interest to assess the influence of neuro-sensitization and F/I balance on dogs quality of life in future studies.

This study represents one of the first steps toward characterizing canine OA pain phenotypes according to neuro-sensitization. Although the small number of healthy dogs was a limitation, the group homogeneity made it possible to distinguish TSP specificity compared to OA dogs. The next steps will be to confirm specificity, validate the inter-rater's reproducibility and determine the sensitivity of response to mechanical TSP and CPM to different treatments in dogs. Although the direct clinical application of QST to assess neuro-sensitization is currently limited by factors such as stress-induced analgesia in clinical settings, future research should aim to develop and validate alternative assessments that are easier to implement in routine practice. Identifying F/I balance will contribute to personalized treatments, by reinforcing inhibitory control, a predictive marker of responsiveness to non-steroidal anti-inflammatory drug in humans [58].

It can be interesting to correlate the neuro-sensitization results with other sensorial changes. Indeed, in human hypersensitivity syndrome, the sensory processing sensitivity (SPS) has been characterized by four components: inhibited behavior, sensitivity to subtle cues, deep information processing, and easy overstimulation [59,60]. Decreased tolerance to auditory, olfactory and visual stimuli has been reported during chronic pain [61–64]. In dogs, the hypersensitivity syndrome has been described though the development of the highly sensitive dog score with ease of excitation, reactivity to environmental stimuli and esthetic sensitivity [65]. Noise phobia was reported for dogs with musculoskeletal impairments (10/10) *vs.* 70% of healthy dogs [66]. Phobia onset was late in the painful dogs, suggesting two different developmental modalities, but this was not discussed in the study. In humans, using functional magnetic resonance imaging (fMRI), the SPS was correlated with specific brain regions such as cingulate and insula, with these regions also contributing to pain

modulation and perception [67,68]. As response to mechanical TSP is a semi-objective assessment, as well as CPM, it would be interesting to obtain fMRI to confirm the regions activated by TSP and CPM [69]. Although trained dogs can be assessed by fMRI without sedation, the pain experienced by OA dogs could bring more challenges to stay in a specific position [70]. A comparative analysis revealed overlaps in many dimensions for the SPS, and the short allele of the serotonin transporter linked polymorphic region (5-HTTLPR) [71]. Interestingly, this short allele contributed to dysfunctional CPM assessed by ischemic pain and PPT, and to decrease inhibitory behavioral control assessed by stop signal task [72,73]. The DNA analysis, particularly for the 5-HTTLPR, would be worth considering to better characterize the dog's hypersensitivity and its possible link to dysfunctional CPM, which would contribute to a pro-nociceptive phenotype.

To conclude, the response to mechanical TSP was translated from cats and validated for its reliability and specificity in dogs. Dogs suffering from OA could develop central sensitization characterized by spinal hyperexcitability. Pro- and anti-nociceptive phenotypes were reported with either an increase in pain endogenous facilitation or inhibition neuromodulation. Better characterization of this F/I imbalance will enable the development of new therapeutic strategies favoring inhibitory controls.

## Acknowledgments

The authors would like to thank all ArthroLab Inc. employees for their technical support, tender care of dogs and their commitment to the welfare of each animal.

## Author contributions

**Conceptualization:** Aliénor Delsart, Maude Barbeau-Grégoire, Aude Castel, Johanne Martel-Pelletier, Jean-Pierre Pelletier, Eric Troncy.

**Data curation:** Aliénor Delsart, Maude Barbeau-Grégoire, Maxim Moreau, Colombe Otis, Eric Troncy.

**Formal analysis:** Aliénor Delsart, Maude Barbeau-Grégoire, Eric Troncy.

**Funding acquisition:** Johanne Martel-Pelletier, Jean-Pierre Pelletier, Eric Troncy.

**Investigation:** Aliénor Delsart, Maude Barbeau-Grégoire, Colombe Otis.

**Methodology:** Aliénor Delsart, Maxim Moreau, Colombe Otis, Aude Castel, Bertrand Lussier, Eric Troncy.

**Project administration:** Maxim Moreau, Johanne Martel-Pelletier, Jean-Pierre Pelletier, Eric Troncy.

**Resources:** Aude Castel, Bertrand Lussier, Johanne Martel-Pelletier, Jean-Pierre Pelletier, Eric Troncy.

**Supervision:** Aude Castel, Johanne Martel-Pelletier, Jean-Pierre Pelletier, Eric Troncy.

**Validation:** Maxim Moreau, Colombe Otis, Eric Troncy.

**Visualization:** Aliénor Delsart.

**Writing – original draft:** Aliénor Delsart, Maude Barbeau-Grégoire, Eric Troncy.

**Writing – review & editing:** Aliénor Delsart, Maude Barbeau-Grégoire, Maxim Moreau, Colombe Otis, Aude Castel, Bertrand Lussier, Johanne Martel-Pelletier, Jean-Pierre Pelletier, Eric Troncy.

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
