## [Decision Letter · Decision Letter 0]

19 Mar 2026

PONE-D-25-54083Evoked temporal summation in dogs to assess pain central sensitization and modulation – A feasibility studyPLOS One

Dear Dr. Troncy,

Thank you for submitting your manuscript to PLOS ONE. After careful consideration, we feel that it has merit but does not fully meet PLOS ONE’s publication criteria as it currently stands. Therefore, we invite you to submit a revised version of the manuscript that addresses the points raised during the review process.

Please address the minor revisions highlighted by the reviewers.

We look forward to receiving your revised manuscript.

Kind regards,

Aliah Faisal Shaheen

Academic Editor

PLOS One

Journal Requirements:

4. Please remove your figures from within your manuscript file, leaving only the individual TIFF/EPS image files, uploaded separately. These will be automatically included in the reviewers’ PDF**.**

5. We note that Figure 1, S1-S5, in your submission contain copyrighted images. All PLOS content is published under the Creative Commons Attribution License (CC BY 4.0), which means that the manuscript, images, and Supporting Information files will be freely available online, and any third party is permitted to access, download, copy, distribute, and use these materials in any way, even commercially, with proper attribution. For more information, see our copyright guidelines: http://journals.plos.org/plosone/s/licenses-and-copyright.

a. You may seek permission from the original copyright holder of Figure 1, S1-S5 to publish the content specifically under the CC BY 4.0 license.

Reviewers' comments:

Reviewer's Responses to Questions

**Comments to the Author**

1. Is the manuscript technically sound, and do the data support the conclusions?

Reviewer #1: Yes

Reviewer #2: Yes

2. Has the statistical analysis been performed appropriately and rigorously? 

Reviewer #1: Yes

Reviewer #2: Yes

3. Have the authors made all data underlying the findings in their manuscript fully available?

Reviewer #1: Yes

Reviewer #2: Yes

4. Is the manuscript presented in an intelligible fashion and written in standard English?

Reviewer #1: Yes

Reviewer #2: Yes

5. Review Comments to the Author

Reviewer #1: I read the manuscript submitted by the authors with great interest and attention. It is an original work that provides relevant data in the field. However, I have raised certain points that require clarification :

Abstract section:

1- Lines 31 and 32: the use of the abbreviation Newton (N) is confusing. I had to wait until the methods section to make the connection. It would be clearer to either specify or write out the term in full.

2- It is unclear why the authors use a threshold of 7% for inhibition functionality. In the methods section, it is reported as ‘according to internal data,’ but I think this should be explained further.

Introduction section:

1- line 61. This manuscript deals with temporal summation; I will not address spatial summation, which is a separate process and has not been tested in the manuscript.

2- Line 70: I am torn about the comments you mentioned. I agree with you that past studies using nociceptive reflexes are not QST. That being said, I do not think that in your study you are measuring the conscious perception of pain and its affective-emotional component, since you are observing behaviour. I would tend to reiterate this point.

Method section:

1- (just out of curiosity) I understand the importance of selecting dogs that obey and respond to commands such as sit or lie down, but with OA, doesn't that immediately exclude overly sensitive dogs that would have been interesting to include in order to have dogs with extreme responses?

2- The protocol has many elements, and I think a diagram that visually represents the two parts of the protocol with the stimulation parameters would be helpful in better understanding and following the sequence of elements.

3- Table 1 - there are more dogs in the tail stimulation protocol than in the ‘forearm’ protocol, so I wonder if the same healthy dogs were used in both protocols. This is not clear in the text.

4- If I understand correctly, the clinical outcome of TSP is the dog's aversive reaction. I think the different reactions should be presented more precisely (in a table, for example) and reported in the results section, as there may be an association between the dog's sensitivity and its reaction.

4 bis - In this regard, I was wondering if the authors had considered taking physiological measurements such as heart rate or breathing rate?

5- Line 194: more information is needed on the 7% threshold.

6- Perhaps I missed it, but was it the same person who performed all the tests for the ICCs?

Results section:

1- There are very few healthy dogs compared to OA dogs. Is there a reason for such a small sample size?

2- Line 276: explain what RMTS is.

Discussion section:

1- Line 416 concerning the fact that dogs are neutered. I may add a section on the role of reproductive hormones in pain perception/modulation.

2- Given that the aim of the project is to determine the feasibility of a TSP protocol in dogs and that several parameters/populations have been tested, I will add a section along the lines of ‘based on our results, here is what we would recommend for measuring TSP in dogs’.

3- I will also mention the potential clinical implications of this protocol. This aspect is glossed over too quickly in the conclusion.

Reviewer #2: I thank the authors for submitting their manuscript. The research addresses an innovative and significant topic for both the journal and clinical practice, promising a substantial impact on enriching current knowledge. This work presents significant advances in the field and highlights a topic of interest to readers. The proposed methodology includes essential elements of the research; however, certain aspects of the manuscript need clarification. I have included some comments that I hope will be useful to the authors.

General comments

L128: Was the quality of life evaluated using a validated scale? If not, please mention this as a limitation.

L199: Please include the analysis of statistical normality and the procedure used for determining the sample size.

6. PLOS authors have the option to publish the peer review history of their article (what does this mean?). If published, this will include your full peer review and any attached files.

Reviewer #1: No

Reviewer #2: **Yes:**Ismael Hernández Avalos

---

## [Author Response · Author response to Decision Letter 1]

17 Apr 2026

Please, see at the attached file: Response to Reviewers 16-04-26.

The authors thank both reviewers for their precious feedbacks and interest in our work. We have carefully addressed all of their comments. Thank you for their valuable suggestions, we believe they have improved the clarity of the manuscript. We hope the revised version meets their expectations.

---

## [Decision Letter · Decision Letter 1]

7 May 2026

Evoked temporal summation in dogs to assess pain central sensitization and modulation – A feasibility study

PONE-D-25-54083R1

Dear Dr. Troncy,

We’re pleased to inform you that your manuscript has been judged scientifically suitable for publication and will be formally accepted for publication once it meets all outstanding technical requirements.

Kind regards,

Aliah Faisal Shaheen

Academic Editor

PLOS One

Additional Editor Comments (optional):

Reviewers' comments:

Reviewer's Responses to Questions

**Comments to the Author**

1. If the authors have adequately addressed your comments raised in a previous round of review and you feel that this manuscript is now acceptable for publication, you may indicate that here to bypass the “Comments to the Author” section, enter your conflict of interest statement in the “Confidential to Editor” section, and submit your "Accept" recommendation.

Reviewer #1: All comments have been addressed

2. Is the manuscript technically sound, and do the data support the conclusions?

Reviewer #1: Yes

3. Has the statistical analysis been performed appropriately and rigorously? 

Reviewer #1: Yes

4. Have the authors made all data underlying the findings in their manuscript fully available?

Reviewer #1: Yes

5. Is the manuscript presented in an intelligible fashion and written in standard English?

Reviewer #1: Yes

6. Review Comments to the Author

Reviewer #1: (No Response)

7. PLOS authors have the option to publish the peer review history of their article (what does this mean?). If published, this will include your full peer review and any attached files.

Reviewer #1: **Yes:**Matthieu Vincenot
